# RicciNet: Deep Clustering via A Riemannian Generative Model

## ABSTRACT

In recent years, deep clustering has achieved encouraging results. However, existing deep clustering methods work with the traditional Euclidean space and thus present deficiency on clustering complex structures. On the contrary, Riemannian geometry provides an elegant framework to model complex structures as well as a powerful tool for clustering, i.e., the Ricci flow. In this paper, we rethink the problem of deep clustering, and introduce the Riemannian geometry to deep clustering for the first time. Deep clustering in Riemannian manifold still faces significant challenges: (1) Ricci flow itself is unaware of cluster membership, (2) Ricci curvature prevents the gradient backpropagation, and (3) learning the flow largely remains open in the manifold. To bridge these gaps, we propose a novel Riemannian generative model (**RICCINET**)[1], a neural Ricci flow with several theoretical guarantees. The novelty is that we model the dynamic self-clustering process of Ricci flow: data points move to the respective clusters in the manifold, influenced by Ricci curvatures. The point's trajectory is characterized by a parametric velocity, taking the form of Ordinary Differential Equation (ODE). Specifically, we encode data points as samples of Guassian mixture in the manifold where we propose two types of reparameterization approaches: Gumbel reparameterization, and geometric trick. We formulate a *differentiable Ricci curvature* parameterized by a Riemannian graph convolution. Thereafter, we propose a geometric learning approach in which we study the geometric regularity of the point's trajectory, and learn the flow via distance matching and velocity matching. Consequently, data points go along *the shortest Ricci flow* to complete clustering. Extensive empirical results show RICCINET outperforms Euclidean deep methods.

## CCS CONCEPTS

• **Computing methodologies** → **Unsupervised learning**; *Neural networks*; • **Information systems** → *Clustering*.

## KEYWORDS

Deep Clustering, Riemannian Geometry, Generative Learning, Ordinary Differential Equation

**ACM Reference Format:**
Anonymous Author(s). 2024. RicciNet: Deep Clustering via A Riemannian Generative Model. In *Proceedings of ACM The Web Conference 2024 (WWW'24)*. ACM, New York, NY, USA, 12 pages. https://doi.org/XXXXXXX.XXXXXXX

---

[1]**Codes are available at** https://anonymous.4open.science/r/RicciNet

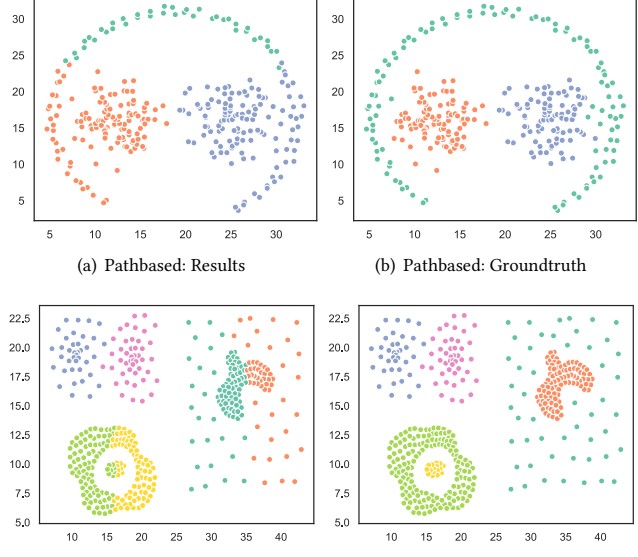

(a) Pathbased: Results     (b) Pathbased: Groundtruth

(c) Compound: Results     (d) Compound: Groundtruth

**Figure 1: Motivated Example. We show the clustering results of a Euclidean deep method, DEC [50]. Groundtruth is on the right. Different colors denote different clusters.**

*Relevance.* Clustering, aiming to group similar samples into the same cluster, is one of the most fundamental tasks in web mining and content analysis. Over the past decades, clustering routinely finds itself in a wide spectrum of applications, ranging from topic discovery of web contents [32] to interest groups mining for recommendation and online advertisement on the World Wide Web [2, 44]. In this paper, we study deep clustering from a fresh perspective of Riemannian geometry, and propose a neural Ricci flow.

## 1 INTRODUCTION

Deep learning methods are becoming the dominant solution for clustering, e.g., variational autoencoders (VAE) [20, 56], generative adversarial nets [33] and the recent contrastive clustering [36]. So far, existing deep clustering methods are in the traditional Euclidean space. Nevertheless, modeling data in Euclidean space is limited. It often falls short of capturing the complexity of real-world scenarios where the data distribution can be highly complicated, especially for the contents on the Web [43, 47]. A primary shortcoming of Euclidean solutions is the deficiency on **clustering complex structures**. For instance, density-based methods usually struggle in segmenting overlapped clusters [53], while VAEs with Gaussian mixture [20] face the difficulty to distinguish the clusters of complex topology. We give a motivated example in Figure 1. A natural question arises: *Is there an effective deep method for more generic clustering, especially for the complex structures?*

In this paper, we study deep clustering from a fundamentally different perspective of Riemannian geometry. **Riemannian geometry** shows better expressiveness for modeling complex structures [13, 26], e.g., hyperbolic space is well aligned with hierarchical structures [42] while hyperspherical space is suitable for cyclic topology

 

[1]. Indeed, Riemannian geometry also provides a powerful tool for clustering, i.e., Ricci flow. Stemming from the thermodynamic equation, Ricci flow demonstrates that particles in the manifold tend to aggregate into several submanifolds, influenced by the Ricci curvature [17]. While the physical phenomenon of Ricci flow aligns with clustering, it has not yet been introduced to deep clustering due to several significant challenges.

The first challenge lies in **modeling cluster membership**. Clustering aims to identify cluster membership of each data sample. On the contrary, Ricci flow itself demonstrates the trend of clustering but does not specify the cluster membership as mentioned above [17]. Also, most flow-based models, also known as normalizing flows, reshape a Gaussian for generation and, as a result, cannot support clustering. The second challenge is **computing Ricci curvature**. The process of Ricci flow is triggered by the Ricci curvature, but computing Ricci curvature is problematic. In the literature, Ricci curvature is well defined with calculating Wasserstein distance between mass distributions [35] or enumerating geodesics triangles in the manifold [12]. The discrete optimization nested within Ricci curvature blocks the gradient backpropagation, thus posing a fundamental challenge to deep clustering. It is not until very recently that a few works [29, 34, 46] investigate on Ricci curvature. Unfortunately, all of them consider the discrete settings, and the formulations cannot be applied to deep clustering. The third challenge is **learning a flow in Riemannian manifold**. Most normalizing flows [38, 39, 49] live in the Euclidean space, and learning the flow is nontrivial. For the discrete flows, the log-determinant of the Jacobian is typically involved and thus results in a cubic complexity, while the recent continuous normalizing flow also requires a costing trace term [22]. In fact, computing distribution density in a Riemannian manifold is rather expensive [42], and it is even more challenging to pushforward the distribution using the flow [30, 31]. We notice that [27, 28] introduce intuitive and efficient ways of flow matching very recently. They focus on data generation in Euclidean space, and are still far from clustering in Riemannian manifold.

To bridge the gaps, we propose a novel neural Ricci flow, **RicciNet**. The novelty is that we model the dynamic self-clustering process of Ricci flow: data points move to the respective clusters in the manifold, influenced by Ricci curvatures. In a nutshell, the trajectory of points' movement is characterized by a parametric velocity, taking the form of Ordinary Differential Equation (ODE), and we analyze geometric regularities of the trajectory to learn the flow. Concretely, for challenge one, we consider a Gaussian mixture in the manifold to identify cluster membership. We propose two types of reparameterization approaches: Gumbel reparameterization and geometric trick. The former is a generative process with wrapped Gaussian in the manifold. The latter is formulated with a linear operation, which is proved to be the manifold-preserving Riemannian operator (Proposition Two). For challenge two, we derive a differentiable formulation with Kantorovich-Rubinstein duality [15], termed as **convolutional Ricci curvature**. It is parameterized by a Riemannian graph convolution on the k-NN graph, modeling the structural information in the meanwhile. Theoretically, we prove that the convolutional Ricci curvature is the upper bound of Ollivier's Ricci curvature (Proposition Three), and thus is regarded as its differentiable alternative. For challenge three, we propose a new

**geometric learning approach**. Instead of explicitly optimizing the likelihood [30, 31], we learn the flow by studying geometric regularity at sample-level thanks to reparameterization. At any time in the process, distances among data points are encouraged to match the ideal distance derived from Ricci flow ODE (distance matching), and the velocity of each point matches the velocity of the shortest path (velocity matching). Consequently, data points go along **the shortest Ricci flow** to the respective clusters.

**Contribution Highlights.** In summary, main contributions are three-fold: **(1) Deep Clustering via Riemannian Manifold.** We rethink the problem of deep clustering and, to the best of our knowledge, make the first attempt to introduce Riemannian geometry to deep clustering for more generic scenario, especially for clustering complex structures. **(2) Neural Ricci Flow.** We propose a novel Riemannian generative RicciNet, a neural Ricci flow with several theoretical guarantees. In particular, we introduce two strategies to reparameterize Gaussian mixture in the manifold, a convolutional Ricci curvature, and a new geometric learning approach to learn the shortest Ricci flow for clustering. **(3) Extensive Experiments.** We evaluate the superiority of RicciNet with 8 strong competitors on 7 datasets, examine the proposed components by ablation study, and further discuss the Ricci flow via visualization.

## 2 PRELIMINARIES

This section first formally reviews the basic concepts of Ricci flow and continuous normalizing flow, and then formulates the studied problem. Important notations are summarized in Appendix A.

### 2.1 Riemannian Geometry

*Riemannian manifold* is a smooth manifold $\mathbb{M}$ endowed with a Riemannian metric $\mathfrak{g}$. Each point $x$ in the manifold is associated with a tangent space $T_x\mathbb{M}$. The transform between manifold and tangent space is done via exponential/logarithmic map, while the transform between two tangent spaces is done via parallel transport. *Constant curvature* $\kappa$ is a global geometric property of the manifold as a whole, and there exists three types of constant curvature manifold: hyperbolic space with negative $\kappa$, hyperspherical space with positive $\kappa$, and Euclidean space, a special case of zero curvature. *Ricci curvature* is a local geometric property of a curve connecting two points in the manifold. A classic definition of Ricci curvature is given by Ollivier [35]. Given two points $x, y$ and mass distributions $m_x, m_y$ surrounding them, Ollivier's formulation is given as

$$Ric(x, y) = 1 - \frac{W_1(m_x, m_y)}{d(x, y)}, \tag{1}$$

where $W_1$ denotes the Wasserstein-1 distance between two distributions, and $d$ is the distance in the manifold. In the literature, Forman gives an alternative definition by enumerating geodesic triangles in the manifold [12]. *Note that, both definitions prevent the gradient backpropagation for deep clustering (Challenge One).* Considering the "heat flow" in the manifold, Hamilton introduces the Ricci flow of a differential equation system that characterizes the self-clustering process [17]. In the thermodynamic system, distances among the points are controlled by Ricci curvature [35],

$$\frac{\partial}{\partial t}d_t(x, y) = -d_t(x, y)Ric(x, y). \tag{2}$$

*Unfortunately, the Ricci flow in Eq. (2) does not specify the cluster membership (Challenge Two).*

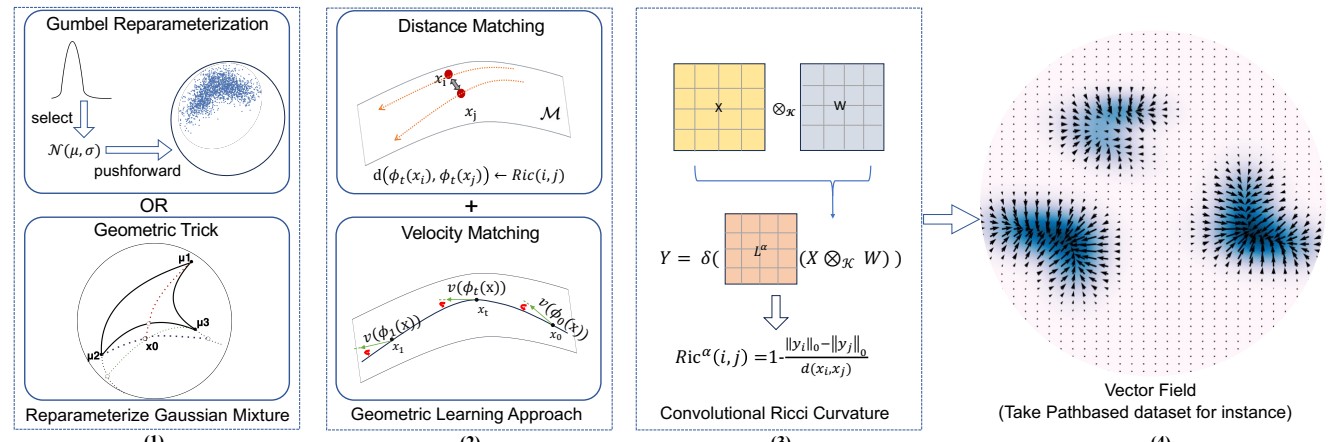

**Figure 2: RicciNet.** We consider a dynamic process in the manifold. In the encoding process, data points move to the respective clusters, and encodings are reparameterized with the parameters of Gaussian mixture in (1) to learn the cluster membership. In the decoding process, points' movement trajectories over time are geometrically regulated by (2), in which a differentiable Ricci curvature in (3) is required to learn the flow. (4) shows the vector field when the points arrive at the Gaussian mixture.

## 2.2 Continuous Normalizing Flow

Continuous normalizing flow (CNF) formulates the vector field of the flow via an Ordinary Differential Equation (ODE) [9], transforming a simple distribution $p_0$ to a complicated one $p_1$. A CNF considers the flow $\phi_t$ as a function over the coordinate of $\boldsymbol{x}$ and time $t \in [0, 1]$, and parameterizes the vector field $v_t$ as a nerual network: $\frac{\partial}{\partial t} \phi_t(\boldsymbol{x}) = v_t(\phi_t(\boldsymbol{x}))$. A probability path $p_t$ is a time-dependent probability density function, i.e., $\int p_t(\boldsymbol{x}) d\boldsymbol{x} = 1, \forall t \in [0, 1]$, and $p_t$ is given by a **pushforward** from $p_0$ for all $t \in [0, 1]$,

$$p_t(\boldsymbol{x}) = [\phi_t]_\star p_0(x) = p_0(\phi_t^{-1}(\boldsymbol{x})) \det \left[ \frac{\partial}{\partial \boldsymbol{x}} \phi_t^{-1}(\boldsymbol{x}) \right], \qquad (3)$$

where $[\phi_t]_\star$ denotes the pushforward along the flow $\phi_t$, det denotes the determinant of a matrix, and $\frac{\partial}{\partial \boldsymbol{x}} \phi_t^{-1}(\boldsymbol{x})$ is the Jacobian matrix. *Learning CNF via Eq. (3) in Euclidean space is nontrivial, and is tougher in Riemannian manifold (Challenge Three).*

## 2.3 Problem Formulation

In this paper, we consider soft clustering. A dataset $\mathcal{D} = \{\boldsymbol{x}_i\}_{i=1}^N$ consists of $N$ unlabelled samples $\boldsymbol{x}_i \in \mathbb{R}^D$ from $K$ clusters. We aim to assign each sample $\boldsymbol{x}_i$ with the cluster membership vector $\boldsymbol{a}_i \in \mathbb{R}^K$. The membership is a stochastic vector adding up to 1, whose $k^{th}$ element is the probability of $\boldsymbol{x}_i$ belonging to cluster $k$.

PROBLEM DEFINITION (DEEP CLUSTERING IN RIEMANNIAN MANIFOLD). *Given the dataset $\mathcal{D}$, the problem is to seek a bijection $\Phi : \boldsymbol{x} \to \boldsymbol{a}$ in the Riemannian manifold $\mathbb{M}$, so that each data point $\boldsymbol{x}$ is mapped to the cluster membership $\boldsymbol{a}$.*

Fundamentally different from existing solutions, we approach deep clustering from a fresh perspective rooted in Riemannian geometry.

## 3 RICCINET: A NEURAL RICCI FLOW

We propose a neural Ricci flow (**RicciNet**), enjoying the expressiveness of a generic manifold and transformation capacity of a flow. Our novelty lies in that, instead of seeking cluster boundary, we study a dynamic self-clustering process, illustrated in Figure 2. In particular, *we model the thermodynamics of Ricci flow: data points move to the respective clusters in the manifold, characterized by Ricci curvatures* as shown in Fig. 2(4). The trajectories of data points is

given by a Riemannian neural ODE (Sec. 3.1). To identify cluster membership, we consider data encodings as samples of a Gaussian mixture, and provide two types of reparameterization tricks in Sec. 3.2. With a differentiable Ricci curvature formulated in Sec. 3.3, we study the geometric regularity of the points' trajectories, and introduce distance/velocity matching to learn the flow (Sec. 3.4).

## 3.1 Riemannian Continuous Normalizing Flow

We consider the data points $\boldsymbol{x}$ in Riemannian manifold, and thus firstly introduce important notions of CNF in the manifold. A *vector field* $v_t$ over the manifold is a smooth function $v_t : [0, 1] \times \mathbb{M} \to T\mathbb{M}$ which maps $(t, \boldsymbol{x})$ to the tangent bundle $T\mathbb{M} = \cup_{x \in \mathbb{M}} \{\boldsymbol{x}\} \times T_x\mathbb{M}$. (In other words, $v_t(\boldsymbol{x})$ lies in the tangent space of $\boldsymbol{x}$.) A *Riemannian probability density* is a nonnegative function $p : \mathbb{M} \to \mathbb{R}_+$ satisfying $\int p(\boldsymbol{x}) d\text{vol}_x = 1$, where $d\text{vol}_x$ is the volume element. A *probability path* is time-dependent and gives the Riemannian density at $t$.

RicciNet models the points' movement trajectories from a base distribution $p_0(x)$ to the observed distribution $p_1(x)$, and the trajectory is described by the velocity vector with an ODE as follows,

$$\frac{\partial}{\partial t} \phi_t(\boldsymbol{x}) = v_t(\phi_t(\boldsymbol{x}); \theta) \in T\mathbb{M}, \quad \phi_0(\boldsymbol{x}) = \boldsymbol{x} \in \mathbb{M}, \qquad (4)$$

$\phi_0(\boldsymbol{x})$ gives the initial state of ODE. In RicciNet, we define $p_0(\boldsymbol{x})$ as the Gaussian mixture in Riemannian manifold,

$$p_0(\boldsymbol{x}) = \sum_k \pi_k \mathcal{N}^{\mathbb{M}}(\boldsymbol{x}|\boldsymbol{\mu}_k, \boldsymbol{\sigma}_k), \qquad (5)$$

where $\pi_k$, $\boldsymbol{\mu}_k$ and $\boldsymbol{\sigma}_k$ denote the mixture coefficient, mean and variance of the $k^{th}$ component Riemannian Gaussian $\mathcal{N}^{\mathbb{M}}$, respectively. Given the probability density path $p_t$ in the Riemannian manifold, $\phi_t$ pushforwards $p_0$ to $p_t = [\phi_t]_\star p_0$, and we have

$$\log(([\phi_t]_\star p_0)(\boldsymbol{x})) = \log(p_0(\boldsymbol{x}')) - \int_0^t \text{div}_g(v_s(\phi_s(\boldsymbol{x}'))) ds, \qquad (6)$$

where $\text{div}_g$ denotes the Riemannian divergence, and $\boldsymbol{x}' = \phi_t^{-1}(\boldsymbol{x})$. A key ingredient of RicciNet is parametric vector field $v_t$. Given the vectors lie in the tangent bundle, $v_t$ is designed as a multilayer perceptron (MLP), whose input layer includes a logarithmic map that project manifold-valued data point to the tangent space.

**Diffeomorphism.** From the perspective of differential geometry, we have the following proposition hold.

PROPOSITION 1 (DIFFEOMORPHISM). *The RICCINET in Eqs. (2) and (3) constructs a diffeomorphism between the Riemannian manifolds of Gaussian mixture and data distribution.*

PROOF. Please refer to Appendix E. □

**$\kappa$-stereographical Model.** We instantiate RICCINET with the $\kappa$-stereographical model $\mathbb{G}_\kappa^d$ [37], as it unifies constant curvature manifold with the gyrovector formalism, and has closed-form expression of Riemannian operators. In particular, the manifold $\mathbb{G}_\kappa^d$ of constant curvature $\kappa$ and dimension $d$ is defined on a smooth gyrovector ball $\left\{ x \in \mathbb{R}^d \mid -\kappa\|x\|^2 < 1 \right\}$ with distance metric of $d(x,y) = \frac{2}{\sqrt{|\kappa|}} \tan_\kappa^{-1}\left( \sqrt{|\kappa|}\| -x \oplus_\kappa y\| \right)$. Gyrovector addition $\oplus_\kappa$, scaling $\otimes_\kappa$, curvature trigonometry e.g. $\tan_\kappa^{-1}$, exponential map $\exp_x^\kappa$, logarithmic map $\log_x^\kappa$, parallel transport $PT_{0\to\mu}^\kappa$ and other operators are summarized in Appendix B. Note that, $\mathbb{G}_\kappa^d$ is hyperspherical with positive $\kappa$, and hyperbolic with negative $\kappa$.

## 3.2 Gaussian Mixture in the Manifold

In RICCINET, we consider a Gaussian mixture as the base distribution to address membership unawareness. In other words, data point $x^1$ will move to and finally arrive at $x^0$ of a Gaussian mixture in the encoding process. We need to rewrite the encoding $x^0$ as a function of Gaussian mixture parameters (mean $\mu_k \in \mathbb{G}_\kappa^d$, covariance $\sigma_k \in \mathbb{R}^d$, and mixture coefficients $\pi \in \mathbb{R}^K$) to learn cluster membership with the mixture. *First*, we derive a soft assignment to each cluster $a$ with solution of Riemannian neural ODE. Given $z = SolveODE(x^1, [0,1], v_t)$, the assignment is defined as $a = Normalize(f(z, \pi, \sigma_1, \cdots, \sigma_K))$, where we use softmax normalizer and $f$ is a neural network. *Second*, we obtain a Gaussian mixture sample $x^0$ from the soft assignment via reparameterization. In particular, we provide two types of novel reparameterization: gumbel reparameterization and geometric trick as in Fig 2(1).

### 3.2.1 *Gumbel Reparameterization.*
We consider the generative process of Gaussian mixture in the manifold: First, a component Gaussian is selected from categorical distribution (Cat), and then a data point is sampled from Riemannian Gaussian. Note that, categorical distribution is problematic as it is not differentiable. To resolve this issue, the first step of our approach is to leverage the differentiable version of Cat (gumbel-softmax [19]) to sharpen $a$,

$$q_i = softmax_{i\in(1,K)}\left( \frac{\log(a_i) + g_i}{\tau} \right), \quad (7)$$

where $\tau$ is a temperature parameter. $g$ is drawn from Gumbel distribution, i.e., $g = -\log(-\log(\epsilon))$ with $\epsilon$ from a uniform distribution, $\epsilon \sim Uniform(0,1)$. Eq. (7) is differentiable over the mixture coefficient. The second step is to instantiate a Riemannian Gaussian via a pushforward in the manifold. In particular, we sample from a standard Gaussian in Euclidean space, $v \sim \mathcal{N}(0,I)$, $v' = \sigma \odot v$, and then conduct the following transform,

$$u = PT_{0\to\mu}^\kappa(v') \in T_0\mathbb{M}, \quad x^0 = \exp_\mu^\kappa(u) \in \mathbb{M}, \quad (8)$$

where $PT^\kappa$ and $\exp^\kappa$ denote parallel transport and exponential map. $\mu$ and $\sigma$ are the mean and covariance of the Gaussian selected by $q$, e.g., $\mu = q^\top[\mu_1, \cdots, \mu_K]^\top$. As a result, the pushforward is given as $h = PT_{0\to\mu}^\kappa \cdot \exp_\mu^\kappa$, yielding the wrapped Gaussian in the manifold, and it is differentiable over the parameters. We provide the detailed algorithm (Algo. 1) and density function in Appendix C and D.

### 3.2.2 *Geometric Trick.*
We follow the geometric intuition that a data point from Gaussian mixture can be expressed as a linear aggregation of the means of component Gaussian in the manifold, and thus uncertainty is given by the weights of aggregation,

$$Linear(\mu_1, \cdots, \mu_K, w) = w[\mu_1, \cdots, \mu_K]^\top \quad (9)$$

where the weight $w$ is a vector-valued function over assignment $a$ and $\epsilon$ uniformly sampled from the ball. Note that, formulating a linear aggregation is challenging in the manifold, owing to the constraint of manifold preserving [37]. Accordingly, we have a stochastic vector $a' = a \odot \epsilon$ and derive the weight function as

$$w(a'|\mu_1, \cdots, \mu_K) = \frac{\lambda_{\mu_i}^\kappa}{\sum_{j=1}^K a_j'(\lambda_{\mu_j}^\kappa - 1)} a', \quad \lambda_\mu^\kappa = \frac{2}{1 + \kappa\|\mu\|^2}, \quad (10)$$

where we use $L_2$ norm, and $\lambda$ is indeed the conformal factor. Consequently, we obtain a differentiable relaxation of Gaussian mixture regarding the parameters of mixture coefficient, mean and variance.

Theoretically, we prove that the formulated linear aggregation is a manifold-preserving Riemannian operator.

PROPOSITION 2 (MANIFOLD PRESERVING). *Given a set of centroids in the manifold $\mu \in \mathbb{G}_\kappa^d$, we have $Linear(\mu_1, \cdots, \mu_K, w) \in \mathbb{G}_\kappa^d$ hold for any set of weights $w \in \mathbb{R}$.*

PROOF. We sketch the proof with key ideas, and present further details in Appendix E. Let $x = Linear(\mu_1, \cdots, \mu_K, w)$. We are to check that $-\kappa\|x\| < 1$ holds, given $\mu \in \mathbb{G}_\kappa^d$. However, tackling inequality is troublesome mathematically, and the equality is preferred. Thus, the key is to leverage the inverse of $\kappa$-stereographical projection $\Gamma$, and investigate the equivalent Lorentz/spherical model of the manifold. (Note that, Lorentz/spherical model $\mathbb{L}_\kappa^d$ is defined on the domain expressed by the equality of $\mathbb{L}_\kappa^d = \{z \in \mathbb{R}^{d+1} | \kappa\langle z, z\rangle_\kappa = 1\}$, where $\langle \cdot, \cdot \rangle_\kappa$ is the metric inner product.) First, applying $\Gamma$, we have $\mu', x' \in \mathbb{L}_\kappa^d$. Second, we check that $\kappa\langle x', x'\rangle_\kappa = 1$ holds for any set of $w \in \mathbb{R}$, completing the proof. □

The geometric trick is able to strictly recover the density of Gaussian mixture in the manifold by training the neural network $f$. However, it is not our focus to recover the density, and more importantly, we will show that the geometric trick achieves competitive and even better clustering results with respect to Gumbel reparameterization.

**Remark 1.** A few studies [18, 49] consider the Discrete NF with Gaussian mixture in the Euclidean counterpart. However, it has not yet been explored on CNF in the manifold to our best knowledge.

## 3.3 Convolutional Ricci Curvature

A Ricci flow is influenced by Ricci curvature. However, its typical definition is given by the discrete optimization and thus blocks the gradient backpropagation, posing a fundamental challenge for deep clustering. To address this challenge, we formulate a differentiable Ricci curvature parameterized by a Riemannian graph convolution, termed as convolutional Ricci curvature, and prove that our formulation is the upper bound of Ollivier's Ricci curvature, and thus is regarded as its differentiable approximation.

### 3.3.1 *Formulation.*
We derive the Ricci curvature on the k-nearest neighbor (k-NN) graph of the data. First, we feed data points collected in $X$ to Riemannian graph convolution on Laplacian $L^\alpha$,

$$Y^\alpha = \delta\left(L^\alpha\left(X \otimes_\kappa W\right)\right), \quad L^\alpha = \alpha I + (1-\alpha)D^{-1}A \quad (11)$$

where $\otimes_\kappa$ denotes $\kappa$-right multiplication, $W$ is the weight matrix,

and $\delta$ is an identity map. $A$ is the adjacency matrix of k-NN graph, and $D$ is the degree matrix of $A$. Then, Ricci curvature $Ric(i, j)$ between two points $x_i$ and $x_j$ is derived as follows,

$$Ric^\alpha(i, j) = 1 - (\|y_i\|_0 - \|y_j\|_0)/d(x_i, x_j), \qquad (12)$$

where the zero-norm is defined as $\|y\| = y\mathbf{1}$, and $d$ is the distance. As a result, our formulation of Ricci curvature in Eq. (12) is differentiable with respect to $x$. Note that, computing the k-NN graph can be boosted by lots of off-the-shelf method [54].

### 3.3.2 Theory.
Here, we elaborate on why our convolutional Ricci curvature is an approximation of Ollivier's Ricci curvature.

PROPOSITION 3 (UPPER BOUND). *The differentiable Ricci curvature in Eq. 12 is the upper bound of Ollivier's Ricci curvature (Eq. 11) in the k-NN graph with the mass distribution given as*

$$m_i^\alpha(x) = \begin{cases} \alpha, & x = i, \\ (1-\alpha)\frac{1}{Degree_i}, & x \in \mathcal{N}_i, \\ 0, & Otherwise, \end{cases} \qquad (13)$$

*where $\mathcal{N}_i$ denotes the neighboring points in the k-NN graph.*

PROOF. Recall Eq. (11). With Kantorovich-Rubinstein duality [15], Wasserstein distance between two distributions is rewritten as

$$W_1(p, q) = \sup_{\|f\|_L \le 1} \mathbb{E}_{z \sim p}[f(z)] - \mathbb{E}_{z \sim q}[f(z)], \qquad (14)$$

where $f$ is 1–Lipschitz. With Eqs. (10), (13) and (15),

$$\begin{aligned} W_1(m_i^\alpha, m_j^\alpha) &= \sup_{\|f\|_L \le 1} \sum_{x \in \mathcal{D}} f(x)m_i^\alpha(x) - \sum_{x \in \mathcal{D}} f(x)m_j^\alpha(x) \\ &= \sup_{\|f\|_L \le 1} [L^\alpha f(X)]_i - [L^\alpha f(X)]_j, \end{aligned} \qquad (15)$$

where $f(X) = (X \otimes_\kappa W)\mathbf{1}$. The operation of $\otimes_\kappa$ is indeed an affine transform [1], and thus $f$ is 1–Lipschitz with proper scaling according to Cauchy-Schwartz inequality. The supremum holds for any feasible $f$, completing the proof. (Details are in Appendix E.) □

Another merit is that our formulation incorporates the structural information for clustering.

## 3.4 Learning the Shortest Ricci Flow

We discuss the challenge of learning the flow in the manifold, and present a fresh idea from a geometric perspective to learn the flow by studying points' movement trajectories in the decoding process.

### 3.4.1 Optimizing the Log-likelihood.
A naïve method of learning Ricci flow is explicitly optimizing the log-likelihood, where we need to specify the probability path. (1) On the one hand, in Riemannian geometry, there exists no closed-form expression of the probability path of Ricci flow, to the best of our knowledge. More importantly, the probability path of original Ricci flow is not related to Gaussian mixture, and thus does not support identifying cluster membership. (2) On the other hand, from the point view of CNF, the probability path is expressed as an integral of Riemannian divergence $\text{div}_\mathfrak{g}$ (Eq. 6). With the Liouville equation [37], we have

$$\text{div}_\mathfrak{g}(v_t(\phi_t(x))) = \sqrt{\det G(\phi_t(x))} tr(\frac{\partial \sqrt{\det G(\phi_t(x))}v_t(\phi_t(x))}{\partial \phi_t(x)}), \qquad (16)$$

where $G(z)$ is the matrix of Riemannian metric. Even though $\kappa$-stereographical model has a closed-form metric ($G(z) = \frac{2}{1+\kappa\|z\|^2}\mathbf{I}_D$ and $\mathbf{I}_D$ is a $D$-dimensional identity matrix), computing the path

is rather complicated and expensive. In particular, computing $\text{div}_\mathfrak{g}$ needs the trace operator where the full Jacobian matrix is required, and it is still nontrivial even with Hutchinson's trace estimator [31]. That is, optimizing the log-likelihood is inferior to learn Ricci flow.

### 3.4.2 A Novel Geometric Approach.
To bridge this gap, we propose a novel geometric approach as in Fig 2 (2). Our insight is to investigate the geometric regularity (i.e., distance and velocity) of movement trajectories throughout the entire decoding process, during which $x^0$ of Gaussian mixture returns to $x^1$ in the manifold. Thanks to the reparameterization proposed in Sec. 3.2, we are able to study the per-sample behavior in the decoding process, instead of the distribution level in terms of probability path. A detailed algorithm of the geometric approach is in Algo. 2 of Appendix C.

**Distance Matching**. We study distances among data points in the decoding process. At any time $t$, the ideal distance among $x^t$ in Ricci flow is specified by the differential equation as follows,

$$\frac{\partial}{\partial t}d(x_i^t, x_j^t) = d(x_i^t, x_j^t)Ric^\alpha(i, j), \qquad (17)$$

where $Ric^\alpha(i, j)$ is given by our differentiable formulation (Sec 3.3). By solving the differential equation, we obtain a close-form solution of the distance over time

$$\hat{d}(x_i^t, x_j^t) = d(x_i^1, x_j^1) \exp((1 - t)Ric^\alpha(i, j)). \qquad (18)$$

Thus, for any two samples of data distribution $p_1$ (saying $x_i^1, x_j^1$), we first obtain the corresponding $x^0$ via the encoding process. Then, during the decoding process, we encourage the distance between them to match the ideal distance given in Eq. (18). Accordingly, the loss of distance matching is formulated as follows

$$\mathcal{L}_{\text{Distance}} = \mathbb{E}_{t, x_i, x_j} \left[ (d(\phi_t(x_i), \phi_t(x_j)) - \hat{d}(x_i^t, x_j^t))^2 \right], \qquad (19)$$

where $\phi_t(x_i) \in \mathbb{M}$ is the location of $x_i$ at time $t$ in the flow, and it is obtained by solving the ODE given in Eq. (4), $t \sim \text{Uniform}(0, 1)$.

**Velocity Matching**. Furthermore, we encourage the data points to go along a shorter path in the neural flow, and we utilize the following fact in Riemannian geometry.

LEMMA (SHORTEST PATH). *In a Riemannian manifold, the shortest path connecting the points $x_0, x_1 \in \mathbb{G}_\kappa^d$ is the geodesics with the curve equation of $x_t = \exp_{x_0}^\kappa(t \log_{x_0}^\kappa(x_1)), t \in [0, 1]$.*

In velocity matching, we suggest that the velocity of the flow from $x^0$ to $x^1$ matches that of geodesics connecting the two points, so as to obtain a shorter and more direct flow. Thus, the loss is given as

$$\mathcal{L}_{\text{Velocity}} = \mathbb{E}_{t, x} \left[ \|v(\phi_t(x); \theta) - \frac{\partial}{\partial t}x_t\|^2 \right], \qquad (20)$$

for any $t \sim \text{Uniform}(0, 1)$ and $x_i \sim p_1$, where we leverage the L2 norm, since velocity vectors are in the Euclidean tangent bundle. The overall loss is formulated with a balancing weight $\beta$ as follows,

$$\mathcal{J}_{\text{RICCINET}} = \mathcal{L}_{\text{Distance}} + \beta \mathcal{L}_{\text{Velocity}}, \qquad (21)$$

The computational complexity to train RICCINET via Eq. (21) is $O(NT|\mathcal{D}|)$, where $|\mathcal{D}|$, $N$ and $T$ denote the size of dataset, number of data samples and number of sampled time points, respectively.

**Remark 2**. CNF conducts an encoding-decoding process, and thus often relates to variational autoencoders (VAE) [22]. Different from VAEs, we regulate the entire decoding process rather than decoding results with Riemannian geometry, so that **data points move to the respective clusters in the manifold along the Ricci flow.**

Table 1: Clustering results on Cora, Citeseer, USPS, MNIST, Reuters, Pathbased, and Compound (denoted as Path and Compo for short) datasets in terms of ACC(%), NMI(%) and ARI(%). The best results are in boldfaced and the runner up underlined.

| Dataset | | DEC [50] | SDCN [4] | DFCN [48] | DEKM [16] | CGC [36] | DRL [53] | ESC [6] | GCF [49] | $\text{RicciNet}_G$ | $\text{RicciNet}_L$ |
|---|---|---|---|---|---|---|---|---|---|---|---|
| Cora | NMI | $41.67_{\pm0.24}$ | $37.38_{\pm0.39}$ | $51.30_{\pm0.41}$ | $25.09_{\pm0.07}$ | $57.03_{\pm0.86}$ | $52.17_{\pm0.13}$ | $21.79_{\pm0.06}$ | $62.10_{\pm1.30}$ | $\mathbf{63.70}_{\pm0.36}$ | $\underline{62.86}_{\pm0.11}$ |
| | ARI | $16.98_{\pm0.29}$ | $13.63_{\pm0.27}$ | $24.46_{\pm0.48}$ | $17.67_{\pm0.13}$ | $49.27_{\pm1.22}$ | $26.91_{\pm1.05}$ | $16.12_{\pm0.02}$ | $63.11_{\pm0.80}$ | $\underline{63.55}_{\pm0.26}$ | $\mathbf{64.20}_{\pm0.40}$ |
| | ACC | $31.92_{\pm0.45}$ | $26.67_{\pm0.40}$ | $37.51_{\pm0.81}$ | $42.39_{\pm0.19}$ | $73.07_{\pm2.05}$ | $19.05_{\pm0.61}$ | $43.95_{\pm0.02}$ | $73.40_{\pm0.63}$ | $\underline{73.95}_{\pm0.28}$ | $\mathbf{75.02}_{\pm0.95}$ |
| Citeseer | NMI | $28.34_{\pm0.30}$ | $38.71_{\pm0.32}$ | $43.90_{\pm0.20}$ | $14.94_{\pm0.03}$ | $44.60_{\pm0.60}$ | $44.23_{\pm0.15}$ | $17.52_{\pm0.05}$ | $40.41_{\pm1.30}$ | $\underline{49.11}_{\pm0.16}$ | $\mathbf{50.02}_{\pm0.52}$ |
| | ARI | $28.12_{\pm0.36}$ | $40.17_{\pm0.43}$ | $45.51_{\pm0.33}$ | $12.54_{\pm0.02}$ | $46.02_{\pm0.55}$ | $15.50_{\pm2.12}$ | $13.02_{\pm0.03}$ | $42.60_{\pm1.21}$ | $\mathbf{48.06}_{\pm1.03}$ | $\underline{47.15}_{\pm0.33}$ |
| | ACC | $55.89_{\pm0.20}$ | $65.96_{\pm0.31}$ | $\mathbf{69.52}_{\pm0.26}$ | $36.80_{\pm0.01}$ | $66.16_{\pm1.20}$ | $20.48_{\pm0.67}$ | $41.75_{\pm0.05}$ | $53.82_{\pm2.20}$ | $67.24_{\pm0.50}$ | $\underline{67.96}_{\pm0.17}$ |
| MNIST | NMI | $77.16_{\pm0.23}$ | $80.90_{\pm0.26}$ | $78.87_{\pm0.34}$ | $89.56_{\pm1.05}$ | $82.21_{\pm0.12}$ | $36.12_{\pm1.60}$ | $86.15_{\pm0.81}$ | $83.07_{\pm0.12}$ | $\underline{91.07}_{\pm1.11}$ | $\mathbf{91.24}_{\pm0.20}$ |
| | ARI | $74.14_{\pm0.18}$ | $72.12_{\pm0.14}$ | $72.62_{\pm0.24}$ | $69.16_{\pm0.08}$ | $71.33_{\pm0.25}$ | $24.56_{\pm0.07}$ | $71.38_{\pm0.56}$ | $67.24_{\pm0.31}$ | $\mathbf{76.52}_{\pm0.49}$ | $\underline{75.80}_{\pm0.22}$ |
| | ACC | $84.30_{\pm0.30}$ | $84.33_{\pm0.23}$ | $84.67_{\pm0.33}$ | $\mathbf{94.65}_{\pm1.30}$ | $87.16_{\pm0.04}$ | $20.27_{\pm3.01}$ | $90.16_{\pm1.13}$ | $85.89_{\pm0.29}$ | $\underline{92.60}_{\pm0.25}$ | $91.56_{\pm1.07}$ |
| USPS | NMI | $65.58_{\pm0.34}$ | $79.15_{\pm0.27}$ | $80.81_{\pm0.30}$ | $78.01_{\pm0.04}$ | $78.15_{\pm0.13}$ | $58.44_{\pm0.14}$ | $69.30_{\pm0.59}$ | $77.52_{\pm0.12}$ | $\mathbf{82.35}_{\pm0.10}$ | $\underline{82.10}_{\pm0.22}$ |
| | ARI | $63.70_{\pm0.27}$ | $71.84_{\pm0.24}$ | $75.30_{\pm0.22}$ | $69.70_{\pm0.04}$ | $74.23_{\pm0.19}$ | $29.07_{\pm0.23}$ | $54.66_{\pm0.56}$ | $70.15_{\pm0.20}$ | $\mathbf{76.63}_{\pm0.24}$ | $\underline{76.59}_{\pm0.63}$ |
| | ACC | $70.71_{\pm0.17}$ | $78.08_{\pm0.19}$ | $79.52_{\pm0.24}$ | $76.93_{\pm0.01}$ | $80.71_{\pm0.11}$ | $18.52_{\pm1.22}$ | $73.64_{\pm0.28}$ | $78.51_{\pm0.60}$ | $\underline{81.02}_{\pm0.16}$ | $\mathbf{81.15}_{\pm0.09}$ |
| Reuters | NMI | $47.50_{\pm0.34}$ | $50.82_{\pm0.21}$ | $59.93_{\pm0.45}$ | $54.13_{\pm0.06}$ | $60.51_{\pm0.15}$ | $32.83_{\pm0.08}$ | $48.25_{\pm0.14}$ | $64.18_{\pm0.33}$ | $\underline{66.35}_{\pm0.13}$ | $\mathbf{67.23}_{\pm0.68}$ |
| | ARI | $48.44_{\pm0.14}$ | $55.36_{\pm0.37}$ | $59.79_{\pm0.36}$ | $59.80_{\pm0.09}$ | $61.62_{\pm0.81}$ | $17.05_{\pm0.63}$ | $49.46_{\pm0.17}$ | $53.12_{\pm1.01}$ | $\underline{63.92}_{\pm0.51}$ | $\mathbf{64.22}_{\pm1.01}$ |
| | ACC | $73.58_{\pm0.13}$ | $77.15_{\pm0.21}$ | $77.70_{\pm0.20}$ | $73.86_{\pm0.02}$ | $77.14_{\pm0.60}$ | $42.28_{\pm1.05}$ | $74.25_{\pm0.05}$ | $78.04_{\pm0.10}$ | $\mathbf{80.25}_{\pm0.85}$ | $\underline{78.95}_{\pm0.20}$ |
| Path | NMI | $34.05_{\pm0.28}$ | $35.65_{\pm3.30}$ | $30.04_{\pm0.15}$ | $30.63_{\pm0.06}$ | $36.05_{\pm0.57}$ | $83.88_{\pm0.10}$ | $34.09_{\pm0.18}$ | $33.15_{\pm0.23}$ | $\underline{84.22}_{\pm0.56}$ | $\mathbf{85.01}_{\pm0.60}$ |
| | ARI | $30.24_{\pm0.12}$ | $29.55_{\pm4.09}$ | $32.35_{\pm2.12}$ | $26.43_{\pm0.13}$ | $26.60_{\pm1.24}$ | $86.93_{\pm1.02}$ | $30.57_{\pm0.23}$ | $24.68_{\pm0.71}$ | $\underline{87.15}_{\pm0.41}$ | $\mathbf{87.20}_{\pm0.22}$ |
| | ACC | $58.33_{\pm0.14}$ | $59.18_{\pm5.74}$ | $59.30_{\pm1.25}$ | $62.55_{\pm0.13}$ | $63.15_{\pm0.93}$ | $21.06_{\pm0.33}$ | $60.95_{\pm0.21}$ | $65.12_{\pm0.60}$ | $\underline{69.03}_{\pm1.17}$ | $\mathbf{71.65}_{\pm0.31}$ |
| Compo | NMI | $54.35_{\pm0.24}$ | $56.76_{\pm1.00}$ | $73.70_{\pm0.72}$ | $43.09_{\pm0.02}$ | $55.23_{\pm1.09}$ | $83.12_{\pm0.16}$ | $34.76_{\pm0.51}$ | $49.10_{\pm0.21}$ | $\underline{85.10}_{\pm0.15}$ | $\mathbf{85.26}_{\pm0.60}$ |
| | ARI | $34.84_{\pm0.14}$ | $37.38_{\pm0.64}$ | $37.83_{\pm0.56}$ | $23.22_{\pm0.09}$ | $31.03_{\pm0.37}$ | $73.63_{\pm0.22}$ | $15.36_{\pm0.23}$ | $36.67_{\pm0.49}$ | $\mathbf{75.27}_{\pm0.23}$ | $\underline{74.51}_{\pm1.02}$ |
| | ACC | $60.62_{\pm0.05}$ | $58.65_{\pm3.14}$ | $59.90_{\pm0.40}$ | $53.29_{\pm0.18}$ | $61.17_{\pm1.10}$ | $28.24_{\pm0.03}$ | $49.15_{\pm3.01}$ | $51.24_{\pm0.10}$ | $\underline{63.51}_{\pm1.02}$ | $\mathbf{65.11}_{\pm0.36}$ |

**Remark 3.** We notice that, very recently, [27] adopts the idea of flow matching in Euclidean space. In Riemannian manifold, [8] pushforwards a Gaussian for generation, while we consider a Gaussian mixture for clustering. In other words, all of them are in different settings to ours, and thus are not applicable to our learning task.

## 4 EXPERIMENT

In this section, we conduct extensive experiment with 8 strong baselines on 7 datasets to (1) evaluate the effectiveness of RicciNet (Sec. 4.2), (2) investigate on the effect of the proposed component of RicciNet (Sec. 4.3), and (3) discuss the curvature of Riemannian manifold and visualize the Ricci flow as a case study (Sec. 4.4).

### 4.1 Experimental Setups

*4.1.1 Datasets.* Without loss of generality, we evaluate our model on a variety of datasets. Concretely, we choose 2 popular image datasets (MINIST and USPS [48]), a text dataset (Reuters [4]), 2 graph datasets (Cora and Citeseer [49]) and 2 challenging artificial datasets of complex structures (Path-based and Compound [53]). The statistics of the datasets is listed in Appendix B.

*4.1.2 Baselines & Evaluation Metrics.* We focus on deep clustering in this paper, and thus we primarily compare our RicciNet with the deep methods. Specifically, we employ 8 strong baselines, including DEC [50], SDCN [4], DFCN [48], DEKM [16], ESC [6], a reinforcement learning model for clustering complex structures (DRL) [53], a contrastive learning model (CGC) [36], and a very recent graph clustering model with normalizing flow (GCF) [49]. Note that, existing deep clustering models are Euclidean. There exists few Riemannian model to the best of our knowledge, and the proposed RicciNet is aimed to bridge this gap. In particular, two instantiations of RicciNet are provided, and RicciNet with Gumbel reparameizeation and geometric trick are referred to as $\text{RicciNet}_G$ and $\text{RicciNet}_L$, respectively. Three popular metrics

are utilized: Normalized Mutual Information (NMI), Adjusted Rand Index (ARI) and Clustering Accuracy (ACC).

*4.1.3 Clustering Euclidean Data.* RicciNet clusters data in the Riemannian manifold. For Euclidean data, we map data points to the manifold, before feeding into RicciNet. One can apply a rescaling to fit the data in the manifold domain [1]. In our design, we opt for applying an exponential map with the reference point of gyrovector ball origin. Thereafter, we employ the matrix $\kappa$-right-multiplication to reduce feature dimension in the manifolds.

*4.1.4 Reproducibility.* To enhance reproducibility, we first specify the neural architecture of RicciNet. The multi-layer perception (MLP) of velocity $v_t$ has three hidden layers. In the reparameterization, $f$ is implemented as another MLP with two hidden layers to output soft assignment. Second, on pretraining and initialization, we suggest to first pretrain the vector field $v_t$ via distance matching at $t = 0$ guided by the ideal Ricci flow, and then initialize the Gaussian mixture accordingly. Third, on parameter configuration, $\alpha$ for Ricci curvature is set as 0.5 following the convention [29, 34]. Scalar curvature of the manifold is a learnable parameter which will be discussed in Sec. 4.4. The means lie in the manifold and thus are optimized by Riemannian Adam [5], while other parameters are optimized by Adam [21]. All the datasets are publicly available. Further details on reproducibility are provided in Appendix F.

*4.1.5 Hardware & Software.* All experiments are done on a serve with GPUs of NVIDIA Tesla V100 and CPUs of Intel i9-10980XE. Our model is built upon PyTorch and GeoOpt [3]. Codes are available at https://anonymous.4open.science/r/RicciNet/

### 4.2 Clustering Results

We show the clustering results in Table 1, and review the motivated example in Fig. 1. As for the clustering results, we conduct 10 independent runs for each model, and report the mean value with

**Table 2: Ablation study on Cora, USPS and Reuters datasets in terms of NMI(%), ARI(%) and ACC(%). The results of the best variants are boldfaced and the runner up underlined.**

| Variant | Cora | | | USPS | | | Reuters | | |
|---|---|---|---|---|---|---|---|---|---|
| | NMI | ARI | ACC | NMI | ARI | ACC | NMI | ARI | ACC |
| $\text{RicciNet}_G$ | 63.70±0.36 | 63.55±0.26 | 73.95±0.28 | 82.35±0.10 | 76.63±0.24 | 81.02±0.16 | 66.35±0.13 | 63.92±0.51 | 80.25±0.85 |
| $\text{RicciNet}_L$ | 62.86±0.11 | 64.20±0.40 | 75.02±0.95 | 82.10±0.22 | 76.59±0.63 | 81.15±0.09 | 67.23±0.68 | 64.22±1.01 | 78.95±0.20 |
| w/oReparameter | **62.15**±0.22 | **62.70**±0.12 | 72.95±0.23 | **82.03**±0.09 | **76.18**±0.08 | **80.96**±0.36 | 65.01±0.50 | 62.92±0.32 | 78.06±0.17 |
| w/oVelocity | 60.82±0.30 | 61.05±0.22 | **73.18**±0.07 | 80.15±0.12 | 75.24±0.33 | 78.11±0.20 | **65.53**±0.13 | 62.06±0.20 | 77.67±0.16 |
| w/oDistance | 58.91±0.10 | 57.33±0.16 | 67.92±0.59 | 78.32±0.11 | 73.52±0.60 | 78.53±0.22 | 62.29±0.36 | 59.33±0.19 | 74.52±0.20 |
| w/oManifold | 58.66±0.31 | 59.70±1.09 | 68.43±0.25 | 76.12±0.18 | 72.01±0.15 | 75.64±0.10 | 60.16±0.19 | 57.03±0.25 | 71.42±0.18 |

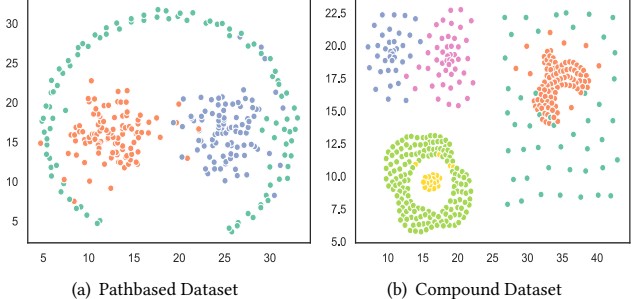

(a) Pathbased Dataset     (b) Compound Dataset

**Figure 3: Visualization of clustering results.**

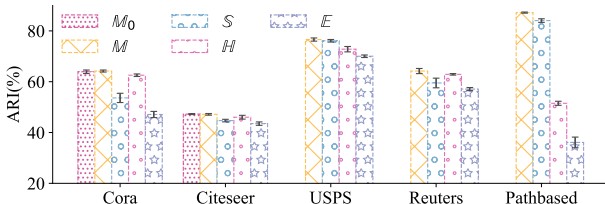

**Figure 4: Clustering results with different curvature in ARI.**

the standard derivation for fair comparison. The empirical results of 8 baselines on all the 7 datasets are summarized in Table 1 in terms of NMI, ARI and ACC. In particular, for graph-based models, we compute the k-NN graph of the dataset in which Euclidean distance is employed to construct the graph. For typical deep clustering models, we neglect the structural information of the graph datasets, i.e., Cora and Citeseer, and utilize the node feature as model input. Note that, our RicciNet consistently achieve the best results in terms of both NMI and ARI, and outperforms the competitors in terms of ACC except two cases.

Next, we zoom in Path-based and Compound datasets of the motivated example in Fig. 1. We visualize the clustering results of $\text{RicciNet}_L$ in Fig. 3, where different clusters are marked by different colors. It shows that our model successfully recovers the cluster structure of the datasets except a few mistakes at the cluster borders. In contrast, clustering results of the advanced deep models are generally undesirable as shown in Table 1 and Fig. 1. Given all of the deep models are Euclidean, we argue that traditional Euclidean metric is limited for clustering, especially for clustering the complex structures. The observations above motivate us to seek for a manifold of better expressiveness, i.e., Riemannian manifold.

### 4.3 Ablation Study

Here, we evaluate the effect of the proposed components of RicciNet: (1) the reparameterization approach, (2) distance matching,

**Table 3: Clustering results of $\text{RicciNet}_L$ with different curvature settings in term of NMI (%).**

| Variant | Cora | Citeseer | USPS | Reuters | Path |
|---|---|---|---|---|---|
| $\mathbb{E}$ | 58.66±0.31 | 42.50±0.67 | 76.12±0.18 | 60.16±0.19 | 39.20±2.04 |
| $\mathbb{H}$ | 61.50±0.11 | 47.16±0.10 | 78.06±0.49 | 65.72±0.21 | 67.18±0.21 |
| $\mathbb{S}$ | 59.11±0.23 | 43.22±0.39 | 81.61±0.15 | 62.91±0.60 | 83.61±0.51 |
| $\mathbb{M}$ | 62.86±0.11 | **50.02**±0.52 | 82.10±0.22 | 67.23±0.68 | 85.01±0.60 |
| $\mathbb{M}_0$ | **63.06**±0.31 | 49.88±0.25 | · | · | · |

(3) velocity matching, and (4) Riemannian manifold. To this end, we design four kinds of variants as follows:

**1) w/oReparameter.** In this variant, we replace the loss of differential geometric learning with the naïve method of optimizing the log-likelihood. The likelihood is computed via Eqs. (6) and (16).

**2) w/oDistance.** To examine the effect of distance matching, we instantiate RicciNet with the reparameterization of geometric trick ($\text{RicciNet}_L$), and train the model by velocity matching loss only.

**3) w/oVelocity.** We disable velocity matching loss of $\text{RicciNet}_L$.

**4) w/oManifold.** To evaluate the effect of introducing Riemannian manifold, we design the proposed model in the Euclidean counterpart. Note that, Ricci flow cannot work for the flat Euclidean space, and thus Euclidean model cannot receive guidance from distance matching. As an alternative, the variant of w/oManifold is designed as the Euclidean version of w/oReparameter, where we leverage the probability path in Euclidean space.

In Table 2, we summarize the clustering results on Cora, USPS and Reuters datasets. (1) Comparing the counterpart variants of w/oReparameter and w/oManifold, it shows that Riemannian model achieves better results than the Euclidean counterpart. A reason is that Riemannian geometry has superior expressiveness to tackle with complex structures [13, 26]. It verifies the motivation of our study, and explains our superiority. (2) Comparing $\text{RicciNet}_G$, $\text{RicciNet}_L$ and w/oReparameter, it shows that re-parameterized model outperforms directly optimizing the likelihood. The reparameterization proposed in Sec. 4.2 involves relaxation, but is shown to be effective for clustering. On the contrary, the probability path in the manifold is grounded on the theory of differential geometry. However, accuracy loss tends to occur in the estimation of Riemannian divergence or integral. (3) Comparing $\text{RicciNet}_L$, w/oDistance and w/oVelocity, we observe that w/oDistance variant consistently has larger performance loss than w/oVelocity except ACC on USPS. It suggests that, *the distance regularity of Ricci flow has the dominant effect on revealing data clusters, which is a key insight of our work.*

### 4.4 Discussion & Visualization

Furthermore, we discuss the effect of constant curvature, and visualize the running example of Path and Compound datasets.

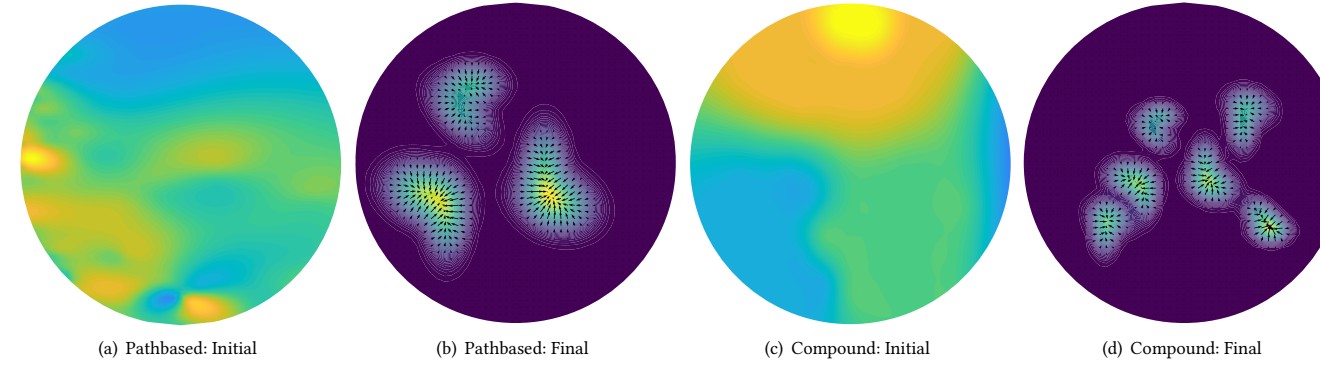

(a) Pathbased: Initial   (b) Pathbased: Final   (c) Compound: Initial   (d) Compound: Final

**Figure 5: Visualization of clustering results.**

To study the constant curvature, we instantiate $\textsc{RicciNet}_L$ with (1) Euclidean space $\mathbb{E}$, (1) standard hyperbolic manifold $\mathbb{H}$ ($\kappa = -1$), (2) standard hyperspherical manifold $\mathbb{S}$ ($\kappa = 1$), (3) a generic manifold $\mathbb{M}$ of learnable curvature, and (4) the manifold $\mathbb{M}_0$ of predefined curvature. For $\mathbb{E}$ variant, we employ the w/oManifold in the ablation study. For $\mathbb{M}_0$ variant, we employ a recent algorithm [13] to estimate the constant curvature for graphs (Cora and Citeseer). Unfortunately, it still remains open to study the constant curvature of data without structural information. (Note that, we cannot estimate curvature with the k-NN graph of pairwise distance, since computing distance also requires the constant curvature). In other words, predefining curvature is not applicable for generic scenario. NMI and ARI of $\mathbb{E}$, $\mathbb{H}$, $\mathbb{S}$, $\mathbb{M}$ and $\mathbb{M}_0$ variants are collected in Table 3 and Fig. 4, respectively. It suggests that it is necessary to fit datasets with learnable curvature as in $\textsc{RicciNet}$. Also, on Cora and Citeseer, we observe that the learnt curvature achieves competitive results to the predefined curvature, explicitly estimated with [13].

As a case study, we visualize the clustering process of $\textsc{RicciNet}_G$ on Path and Compound datasets. In particular, the (gyrovector) manifold is set to as $2D$-ball for the ease of visualization, and the constant curvature is jointly learnt with the model. We run $\textsc{RicciNet}_G$, and plot data points in balls where the lighter color represents the denser data distribution. Taking Path dataset for instance, Fig. 5 (a) is the initial ball showing the original data distribution on the manifold, while Fig 5 (b) shows the clustering results at the final state. As shown in Fig. 5, *data points flow to the respective clusters on the manifold, according to the guidance of our neural Ricci flow.*

## 5 RELATED WORK

***Deep Clustering.*** Clustering is unsupervised by nature, and thus deep clustering frequently revisits neural architectures as follows: (1) autoencoder [50], (2) variational autoencoder (VAE) [20, 24], and (3) generative adversarial nets (GAN) [33]. (4) Graph neural networks (GNN) are leveraged to capture the structural information of the data for boosting clustering performances [4]. (5) Contrastive clustering explores the similarity of the data themselves, and is receiving increasing attention recently [25, 36]. (6) As for the normalizing flow (NF), some consider a variational mixture of flows [38], while others study the flow based on Gaussian mixture for clustering [18]. A more detailed survey is given in [55]. Very recently, DRL [53] introduces reinforcement learning to density based clustering. GCF [49] is presented as a discrete NF on Euclidean space for clustering graph data. In contrast, we study the continuous NF on

the manifold for generic clustering. To the best of our knowledge, existing deep methods lie in Euclidean space, and we are the first to introduce Riemannian geometry to deep clustering.

***Riemannian Machine Learning.*** Euclidean space has been the workhorse for machine learning for decades, and Riemannian manifolds emerge as an exciting alternative, e.g., hyperbolic space shows superiority in hierarchical structures [52], while hyperspherical space is suitable for cyclic ones [1]. In recent years, researchers investigate various manifold types [23, 51] and neural architectures [14, 41], and successfully conduct classification on texts, images and graphs [7, 45]. Surprisingly, clustering has been rarely explored in the manifolds. In the literature, [10] extends a variant of k-means on the manifold. [11] optimizes over a matrix manifold for clustering graph data specially. None of them consider deep clustering for general purpose. Also, we notice that Ricci curvature is receiving research attention recently, and [29, 34, 46] introduce Ricci curvature to address the over-squashing of graph neural networks. On the contrary, Ricci flow is still under explored yet, and we make an attempt to design a neural Ricci flow for clustering.

***Continuous Normalizing Flow.*** Normalizing flow is a family of generative methods that reshape data distribution through a series of invertible mappings [39], and we focus on the continuous normalizing flow (CNF) in this paper. The vast majority of CNFs work with Euclidean space [22], and it is not until recently that a few CNFs are designed in Riemannian manifold. Concretely, [30, 31] study the probability path in the manifold, while [8, 40] present geometric methods to learn the flow. However, they focus on pushforwarding a Gaussian for data generation, while we consider a Gaussian mixture on the manifold for clustering.

## 6 CONCLUSION

In this paper, we study deep clustering from a fundamentally different perspective of Riemannian geometry, and propose a novel generative neural Ricci flow ($\textsc{RicciNet}$), which bridges the data observations and a Gaussian mixture for clustering. In particular, we encode data point as a sample of Gaussian mixture in which we propose two types of reparameterization approaches. In the whole decoding process, per-sample behavior is geometrically regulated by velocity matching and distance matching based on differentiable Ricci curvature, which is formulated as a Riemannian graph convolution. As a result, the data points move to the respective clusters on the manifold along the shortest Ricci flow. Extensive empirical results show the superiority of $\textsc{RicciNet}$ on a variety of datasets.

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

**Table 1: Glossary of Important Notations**

| Symbol | Description |
|---|---|
| $\mathbb{H}, \mathbb{S}, \mathbb{M}$ | Hyperbolic, hyperspherical and generic manifold |
| $\mathbb{G}_\kappa^d$ | $\kappa$−stereographical model of Riemannian manifold |
| $\kappa, d$ | Constant curvature and dimension, respectively |
| $\lambda_{\boldsymbol{x}}^\kappa$ | Conformal factor of the point $\boldsymbol{x}$ in manifold $\mathbb{G}_\kappa^d$ |
| $Ric(\boldsymbol{x}, \boldsymbol{y})$ | Ricci curvature between points $\boldsymbol{x}$ and $\boldsymbol{y}$ |
| $\mathcal{N}^{\mathbb{M}}$ | Gaussian distribution in Riemannian manifold |
| $\boldsymbol{\nu}_k, \boldsymbol{\sigma}_k$ | The mean in the manifold and covariance of $\mathcal{N}^{\mathbb{M}}$ |
| $\boldsymbol{\pi}$ | Mixture coefficients of the Gaussian mixture |
| Uniform(0, 1) | Uniform distribution |
| $T_{\boldsymbol{x}}\mathbb{M}, T\mathbb{M}$ | Tangent space of $\boldsymbol{x}$, Tangent bundle |
| $p_t$ | Probability path in the manifold, $t \in [0, 1]$ |
| $\phi_t(\boldsymbol{x})$ | Flow of the point $\boldsymbol{x}$, $\phi_t(\boldsymbol{x}) \in \mathbb{M}$ |
| $v_t(\phi_t(\boldsymbol{x}); \theta)$ | Parametric vector field, $v_t(\phi_t(\boldsymbol{x}); \theta) \in T\mathbb{M}$ |

**Table 2: Statistics of datasets**

| Data | Type | Datapoint | Cluster | Feature | Link |
|---|---|---|---|---|---|
| Cora | Graph | 2708 | 7 | 1433 | 5429 |
| Citeseer | Graph | 3327 | 6 | 3703 | 4732 |
| MNIST | Image | 70000 | 10 | 28×28 | - |
| USPS | Image | 9298 | 10 | 16×16 | - |
| Reuters | Text | 10000 | 4 | 2000 | - |
| Path | Artificial | 300 | 3 | 2 | - |
| Compound | Artificial | 788 | 7 | 2 | - |

## A   NOTATIONS

We summarize important notations of this paper in Table 1.

## B   DATASETS & BASELINES

To evaluate our model, we choose 7 datasets of texts, images and graphs, and the statistics are detailed in Table 2. We include 8 strong baselines, introduced as follows,

- **DEC** [50] trains an autoencoder to learn the deep representations for clustering.
- **SDCN** [4] considers structural information among the data by integrating a GCN to an autoencoder.
- **DFCN** [48] is equipped with a structure and attribute information fusion (SALF) module for boosting clustering.
- **DEKM** [16] alternately optimizes representation learning and clustering via a greedy method.
- **CGC** [36] conducts contrastive learning at different levels for end-to-end graph clustering[2].
- **DRL** [53] learns the optimal search strategy of clustering parameters for data distributions via reinforcement learning.
- **ESC** [6] analyzes the special behavior of Wasserstein center of gravity in clustering probability distribution and proposes a distance-based K-means algorithm.
- **GCF** [49] integrates GCN and the discrete NF based on Gaussian mixture for graph clustering.

Note that, CGC and GCF are originally designed for clustering graph data, and thus we apply them on a k-NN graph of the data. Existing deep clustering methods work with Euclidean space.

---

[2]The static version of CGC is included for comparison as the datasets do not provide temporal information

---

**Algorithm 1** Gumbel Reparameterization of Gaussian Mixture in the Manifold

**Input:** Soft assignment $\boldsymbol{\sigma}$ of a sample $\boldsymbol{x}^0$, $K$ Gaussian components with mean $\boldsymbol{\mu} \in \mathbb{M}$ and covariance $\boldsymbol{\sigma}$;
**Output:** Rewritten $\boldsymbol{x}$ with parameters of the Gaussian mixture;
1: Sample $\epsilon \sim$ Uniform(0, 1);
2: Compute $g = -\log(-\log(\epsilon))$;
3: $\mathbf{q}$ = GumbelSoftmax($\boldsymbol{a}, g$);
4: Select a component Gaussian with $\boldsymbol{\mu}, \boldsymbol{\sigma}$ via category $\mathbf{q}$;
5: Sample $\boldsymbol{v} \sim \mathcal{N}(\boldsymbol{0}, \mathbf{I})$ in Euclidean space;
6: Scale $\boldsymbol{v}$ according to the covariance: $\boldsymbol{v}' = \boldsymbol{\sigma} \odot \boldsymbol{v}$;
7: Parallel transport $\boldsymbol{v}'$ to $\boldsymbol{u}$ in the tangent space of the mean;
8: Project $\boldsymbol{u}$ to the manifold via exponential map $\exp_{\boldsymbol{\mu}}^\kappa$;
9: **return** Reparameterized $\boldsymbol{x}^0$ as a differentiable function over mixture coefficient, mean and covariance.

---

**Algorithm 2** Procedure of Geometrically Learning RICCINET

**Input:** Dataset $\mathcal{X}$ with optional structure $\mathbf{G}$, Parametric vector field $v_t$, The number of clusters $K$; The type of reparameterization;
**Output:** Parameters of RICCINET;
1: **if** Graph data **then**
2: $\quad$ $\mathbf{A} = \mathbf{G}$ and do not compute k-NN graph;
3: **else**
4: $\quad$ $\mathbf{A} =$ kNN-Graph($\mathcal{X}$) with distance metric in the manifold;
5: **end if**
6: Pretain $v_t$ and initialize Gaussian mixture;
7: **while** not converged **do**
8: $\quad$ **for** each data point in $\mathcal{X}$ **do**
9: $\quad\quad$ Obtain $N$ samples for each data point;
10: $\quad\quad$ Encode a sample by $SolveODE(\boldsymbol{x}^1, [0, 1], v_t)$;
11: $\quad\quad$ Compute the soft assignment $\boldsymbol{a}$ of the sample;
12: $\quad\quad$ **if** Gumbel Reparameterization **then**
13: $\quad\quad\quad$ Call Algo. 1 to obtain reparameterized $\boldsymbol{x}^0$;
14: $\quad\quad$ **else**
15: $\quad\quad\quad$ Obtain reparameterized $\boldsymbol{x}^0$ via a linear operator;
16: $\quad\quad$ **end if**
17: $\quad\quad$ Sample a set of time points $t \sim$ Uniform(0, 1);
18: $\quad\quad$ Decode $\boldsymbol{x}^0$ and obtain $\boldsymbol{x}^t = SolveODE(\boldsymbol{x}^0, [0, t], v_t)$;
19: $\quad\quad$ Derive convolutional Ricci curvature with $\mathbf{A}$;
20: $\quad\quad$ Compute ideal distance $\hat{d}$ of Ricci flow;
21: $\quad\quad$ Compute distance between $\boldsymbol{x}^t$, and $\mathcal{L}_{\text{Distance}}$;
22: $\quad\quad$ Derive the geodesics connecting $\boldsymbol{x}^0$ and $\boldsymbol{x}^1$;
23: $\quad\quad$ Compute partial derivative of the geodesics;
24: $\quad\quad$ Compute $v_t$ and $\mathcal{L}_{\text{Velocity}}$;
25: $\quad\quad$ Optimize parameters by min $\mathcal{L}_{\text{Distance}} + \beta\mathcal{L}_{\text{Velocity}}$
26: $\quad$ **end for**
27: **end while**

## C   ALGORITHMS

The proposed Gumbel reparameterization is summarized in Algo. 1. The overall procedure of our geometric learning approach is given in Algo. 2. The computational complexity of Algo. 2 is $O(NT|\mathcal{X}|)$, where $|\mathcal{X}|$, $N$ and $T$ denote the size of dataset, number of data samples and number of sampled time points, respectively. Computing k-NN graph is a pre-processing, and can be boosted via [54]. In Line

**Table 3: Summary of the operations with unified formalism.**

| Operation | Unified gyrovector formalism in $\mathbb{G}_\kappa^d$ | Euclidean counterpart |
|---|---|---|
| Distance Metric | $d_{\mathcal{M}}^\kappa(\mathbf{x}, \mathbf{y}) = \frac{2}{\sqrt{|\kappa|}} \tan_\kappa^{-1}\left(\sqrt{|\kappa|} \, \|-\mathbf{x} \oplus_\kappa \mathbf{y}\|_2\right)$ | $d(\mathbf{x}, \mathbf{y}) = \|\mathbf{x} - \mathbf{y}\|_2$ |
| Gyrovector Addition | $\mathbf{x} \oplus_\kappa \mathbf{y} = \frac{(1 - 2\kappa\langle\mathbf{x},\mathbf{y}\rangle - \kappa\|\mathbf{y}\|_2^2)\mathbf{x} + (1 + \kappa\|\mathbf{x}\|_2^2)\mathbf{y}}{1 - 2\kappa\langle\mathbf{x},\mathbf{y}\rangle + \kappa^2\|\mathbf{x}\|_2^2\|\mathbf{y}\|_2^2}$ | $\mathbf{x} \oplus_\kappa \mathbf{y} = \mathbf{x} + \mathbf{y}$ |
| Gyrovector Scaling | $r \otimes_\kappa \mathbf{x} = \frac{1}{\sqrt{\kappa}} \tanh\left(\kappa \tanh^{-1}(\sqrt{\kappa}\|\mathbf{x}\|_2)\right) \frac{\mathbf{x}}{\|\mathbf{x}\|_2}$ | $r \otimes_\kappa \mathbf{x} = r\mathbf{x}$ |
| Matrix-Vector Multiplication | $\mathbf{M} \otimes_\kappa \mathbf{x} = (1/\sqrt{\kappa}) \tanh\left(\frac{\|\mathbf{M}\mathbf{x}\|_2}{\|\mathbf{x}\|_2} \tanh^{-1}(\sqrt{\kappa}\|\mathbf{x}\|_2)\right) \frac{\mathbf{M}\mathbf{x}}{\|\mathbf{M}\mathbf{x}\|_2}$ | $\mathbf{M} \otimes_\kappa \mathbf{x} = \mathbf{M}\mathbf{x}$ |
| $\kappa$-Right-Multiplication | $X \otimes_\kappa W = \exp_0^\kappa(\log_0^\kappa(X)W)$ | $X \otimes_\kappa W = XW$ |
| Exponential Map | $\exp_\mathbf{x}^\kappa(\mathbf{v}) = \mathbf{x} \oplus_\kappa \left(\tan_\kappa\left(\sqrt{|\kappa|}\frac{\lambda_\mathbf{x}^\kappa\|\mathbf{v}\|_2}{2}\right)\frac{\mathbf{v}}{\|\mathbf{v}\|_2}\right)$ | $\exp_\mathbf{x}^\kappa(\mathbf{v}) = \mathbf{x} + \mathbf{v}$ |
| Logarithmic Map | $\log_\mathbf{x}^\kappa(\mathbf{y}) = \frac{2}{\lambda_\mathbf{x}^\kappa\sqrt{|\kappa|}} \tan_\kappa^{-1} \|-\mathbf{x} \oplus_\kappa \mathbf{y}\|_2 \frac{-\mathbf{x} \oplus_\kappa \mathbf{y}}{\|-\mathbf{x}\oplus_\kappa\mathbf{y}\|_2}$ | $\log_\mathbf{x}^\kappa(\mathbf{y}) = \mathbf{x} - \mathbf{y}$ |
| Parallel Transport | $PT_{\mathbf{x}\to\mathbf{y}}^\kappa(\mathbf{v}) = -\frac{\lambda_\mathbf{x}^\kappa}{\lambda_\mathbf{y}^\kappa}(\mathbf{y} \oplus_\kappa -\mathbf{x}) \oplus_\kappa (\mathbf{y} \oplus_\kappa (-\mathbf{x} \oplus_\kappa \mathbf{v}))$ | $PT_{\mathbf{x}\to\mathbf{y}}^\kappa(\mathbf{v}) = \mathbf{v} - \mathbf{x} + \mathbf{y}$ |
| Curvature Trigonometry | $\tan_\kappa(\mathbf{x}) = \begin{cases} \tanh(\mathbf{x}), & \kappa < 0, \\ \tan(\mathbf{x}), & \kappa > 0. \end{cases}$ $\cos_\kappa(\mathbf{x}) = \begin{cases} \cosh(\mathbf{x}), & \kappa < 0, \\ \cos(\mathbf{x}), & \kappa > 0. \end{cases}$ $\sin_\kappa(\mathbf{x}) = \begin{cases} \sinh(\mathbf{x}), & \kappa < 0, \\ \sin(\mathbf{x}), & \kappa > 0. \end{cases}$ | $\tan(\mathbf{x})$ $\cos(\mathbf{x})$ $\sin(\mathbf{x})$ |
| Applying Function | $f^{\otimes_\kappa}(\mathbf{x}) = \exp_\mathbf{o}^\kappa\left(f\left(\log_\mathbf{o}^\kappa(\mathbf{x})\right)\right)$ | $f(\mathbf{x})$ |

6 of Algo. 2, we suggest to pretrain the vector field $v_t$ via distance matching. In particular, we have $\boldsymbol{x}^0 = SolveODE(\boldsymbol{x}^1, [0, 1], v_t)$ by solving the ODE. We first pretrian $v_t$ by matching the distance among $\boldsymbol{x}^0$ to that given by ideal Ricci flow at $t = 0$, and then initialize the parameters of Gaussian mixture accordingly. Note that, solving ODEs as well as backpropagating the gradient is well studied [9], and the ODE is endowed with Riemannian manifold via exponential/logarithmic map. All Riemannian operators are closed-formed, and given in the next Sec.

## D  RIEMANNIAN GEOMETRY

We formally review the operators in the manifold, and specify the density of wrapped Gaussian.

A Riemannian manifold $(\mathcal{M}, g)$ is a smooth manifold $\mathcal{M}$ endowed with a Riemannian metric $g$. Every point $\mathbf{x} \in \mathcal{M}$ is associated with a Euclidean-like *tangent space* $\mathcal{T}_\mathbf{x}\mathcal{M}$ on which the metric $g$ is defined to shape the manifold. The collection of tangent spaces over the manifold is said to be *tangent bundle*, denoted as $\mathcal{T}\mathcal{M}$. Given a point in the manifold $\mathbf{x} \in \mathcal{M}$, the *exponential map* projects a vector $\mathbf{v}$ in the tangent space at $\mathbf{x}$ to the manifold $exp_\mathbf{x}(\mathbf{v}) : \mathcal{T}_\mathbf{x}\mathcal{M} \to \mathcal{M}$. The *logarithmic map* projects a point $\mathbf{y}$ in the manifold to the tangent space of $\mathbf{x}$, $log_\mathbf{x}(\mathbf{y}) : \mathcal{M} \to \mathcal{T}_\mathbf{x}\mathcal{M}$, serving as the inverse of exponential map. Both exponential and logarithmic maps are locally defined with a reference point $\mathbf{x}$. The *parallel transport* carries the vector in one tangent space to another along the geodesic $PT_{\mathbf{x}\to\mathbf{y}}(\mathbf{v}) : \mathcal{T}_\mathbf{x}\mathcal{M} \to \mathcal{T}_\mathbf{y}\mathcal{M}$. The *geodesic* is the shortest curve connecting two points in the manifold. In particular, given the curve as the function of manifold-valued coordinates with respect to time $\boldsymbol{x}_t : [a, b] \to \mathcal{M}$, the geodesic is found by solving the optimization of $\boldsymbol{x}_t = \arg\min_{\boldsymbol{x}_t} \frac{1}{2} \int_b^a \boldsymbol{x}_t'^\top G(\boldsymbol{x}_t')\boldsymbol{x}_t' dt$, where $G(\boldsymbol{x}_t')$ is the matrix of Riemannian metric. $\boldsymbol{x}_t'$, the first-order derivative, is the velocity of $\boldsymbol{x}_t$ lying in the tangent space of $\boldsymbol{x}_t$. The integral in the optimization is indeed the square of curve length. With the

notions above, we have the description as follows. *In RicciNet, the Flow of a point is curve of a point's trajectory in the manifold, and is characterized by an ODE endowed with manifold metric. Thus, the velocity/vector field of the flow lies in the tangent bundle of the manifold.* (For further facts on Riemannian geometry, refer to [37].)

In the literature, there are several model to work with Riemannian manifold, such as Klein model, $\kappa$-stereographical model and Lorentz model [1, 41]. In RicciNet, we opt for $\kappa$-stereographical model in which the exponential map, logarithmic map and parallel transport has closed-form expression. In addition, $\kappa$-stereographical model is defined on a gyrovector space, and thus supports Euclidean-like vector operations, such as addition, scaling and matrix-vector multiplication. We summarize the important operators in Table 3. Gyrovector operations converges to the Euclidean counterpart in the limit of zero curvature. On curvature trigonometry, $\arcsin_\kappa$, $\arccos_\kappa$, and $\arctan_\kappa$ are curvature aware as $\sin_\kappa$, $\cos_\kappa$, and $\tan_\kappa$.

In the manifold, a wrapped Gaussian is given by Line 5-8 in Algo. 2. Accordingly, the density of wrapped Gaussian $\mathcal{N}^{\mathbb{M}}$ is derived by a pushforward $f$ from a standard Gaussian $\mathcal{N}$.

$$\log \mathcal{N}^{\mathbb{M}}(\boldsymbol{z}; \boldsymbol{\mu}, \boldsymbol{\sigma}) = \log \mathcal{N}(\boldsymbol{v}; \boldsymbol{\mu}_0, \boldsymbol{\sigma}) - \log\det(\frac{\partial f}{\partial \boldsymbol{v}}), \quad (1)$$

where det denotes the determinant and $\frac{\partial f}{\partial \boldsymbol{v}}$ is the Jacobian. In particular, the pushforward is given as $f = \exp_{\boldsymbol{\mu}} \circ PT_{\boldsymbol{\mu}_0 \to \boldsymbol{\mu}}$ with is the reverse procedure of

$$\boldsymbol{u} = \log_{\boldsymbol{\mu}}(\boldsymbol{z}) \in \mathcal{T}_{\boldsymbol{\mu}}\mathcal{M}, \quad \boldsymbol{v} = PT_{\boldsymbol{\mu}\to\boldsymbol{\mu}_0}(\boldsymbol{u}) \in \mathcal{T}_{\boldsymbol{\mu}_0}\mathcal{M}. \quad (2)$$

In $\kappa$-stereographical model, we are able to get a clean closed-form determinant of the Jacobian. Another important property is that the density of Eq. (1) converges the that of standard Gaussian when the constant curvature approaches to zero [42].

## E  PROOFS

Here, we detail the proofs of the three propositions proposed in this paper. We rewrite the propositions to be self-contained.

PROPOSITION 1 (DIFFEOMORPHISM). *The RICCINET in Eqs. (2) and (3) constructs a diffeomorphism between the Riemannian manifolds of Gaussian mixture and data distribution.*

PROOF. First, we introduce the definition of diffeomorphism in differential geometry. Given two manifolds $\mathcal{M}_0$ and $\mathcal{M}_1$, a smooth map $\varphi : \mathcal{M}_0 \to \mathcal{M}_1$ is referred to as a diffeomorphism if $\varphi$ is bijective and its inverse $\varphi^{-1}$ is also smooth. $\mathcal{M}_0$ and $\mathcal{M}_1$ are said to be diffeomorphic and denoted as $\mathcal{M}_0 \simeq \mathcal{M}_1$ if there exists a $\varphi$. In other words, the proposition holds if and only if there exists a bijection $\varphi$ and $\varphi^{-1}$. Second, we specify the invertible bijection implicitly given in the ODE of RICCINET. It takes the form as follows,

$$\varphi(\boldsymbol{x}^0) = SolveODE(\boldsymbol{x}^1, [0, 1], v_t) \in \mathcal{M}_1, \quad (3)$$

$$\varphi^{-1}(\boldsymbol{x}^1) = SolveODE(\boldsymbol{x}^0, [0, 1], v_t) \in \mathcal{M}_0. \quad (4)$$

The existence of invertible bijection is in accordance with the design of normalizing flow. That is, the ODE gives a diffeomorphism onto the manifold itself, connecting two type of structures. □

PROPOSITION 2 (MANIFOLD PRESERVING). *Given a set of centroids in the manifold $\boldsymbol{\mu} \in \mathbb{G}_\kappa^d$, we have $\mathrm{Linear}(\boldsymbol{\mu}_1, \cdots, \boldsymbol{\mu}_K, \boldsymbol{w}) \in \mathbb{G}_\kappa^d$ hold for any set of weights $w \in \mathbb{R}$.*

PROOF. The proof involves heavy algebra. We give the key equations to support the proof, instead of buried in the algebra. First, we introduce the Lorentz/spherical model, which connects to $\kappa$-stereographical model with stereographic projection $\Gamma$. Concretely, Lorentz/spherical model is defined in $\mathbb{L}_\kappa^d = \{z \in \mathbb{R}^{d+1} | \kappa \langle z, z \rangle_\kappa = 1\}$, where hyperbolic and hyperspherical spaces are unified in the curvature-aware metric inner product

$$\langle z, z \rangle_\kappa = sgn(\kappa) z_t^2 + z_s^\top z_s, \quad \forall z = [z_t \ z_s]^\top \in \mathbb{L}_\kappa^d, \quad (5)$$

where $sgn$ is the sign function. $z$ is rewritten as the time-space coordinates. The projection is given as

$$\Gamma([z_t \ z_s]^\top) = \frac{1}{1 + \sqrt{|\kappa|} z_t} z_s \to \boldsymbol{x} \in \mathbb{G}_\kappa^d \quad (6)$$

$$\Gamma^{-1}(\boldsymbol{x}) = \left( \frac{1}{\sqrt{|\kappa|}} (\lambda_{\boldsymbol{x}}^\kappa - 1), \lambda_{\boldsymbol{x}}^\kappa \boldsymbol{x} \right) \to z \in \mathbb{L}_\kappa^d, \quad (7)$$

where $\lambda_{\boldsymbol{x}}^\kappa$ is the conformal factor. Stereographic projection gives a perfect duality of gyrovector ball and Lorentz model. Then, we have the proposition hold if and only if the following equality

$$\frac{1}{\kappa} (\lambda_{\boldsymbol{x}}^\kappa - 1)^2 + (\lambda_{\boldsymbol{x}}^\kappa)^2 \boldsymbol{x}^\top \boldsymbol{x} = \frac{1}{\kappa}, \quad (8)$$

is ensured with $\boldsymbol{\mu} \in \mathbb{G}_\kappa^d$. Indeed, Eq. (8) is verified. Alternatively, one can have a quick check by investigating the gyro-midpoint as

$$\mathrm{mid}_\kappa(\boldsymbol{x}_1, \cdots, \boldsymbol{x}_K; \boldsymbol{\alpha}) = \frac{1}{2} \otimes_\kappa \left( \sum_{i=1}^n \frac{\alpha_i \lambda_{\boldsymbol{x}_i}^\kappa}{\sum_{j=1}^n \alpha_j (\lambda_{\boldsymbol{x}_j}^\kappa - 1)} \boldsymbol{x}_i \right), \quad (9)$$

where $\boldsymbol{\alpha}$ is the vector collecting the weights. As the midpoint lies in the manifold, the re-scaled midpoint is also manifold preserving. Note that, $\boldsymbol{x} \in \mathbb{G}_\kappa^d$ holds for any $\frac{1}{2} \otimes \boldsymbol{x} \in \mathbb{G}_\kappa^d$. □

PROPOSITION 3 (UPPER BOUND). *The differentiable Ricci curvature in Eq. 12 is the upper bound of Ollivier's Ricci curvature (Eq. 11) in the k-NN graph with the mass distribution given as*

$$m_i^\alpha(x) = \begin{cases} \alpha, & x = i, \\ (1-\alpha) \frac{1}{Degree_i}, & x \in \mathcal{N}_i, \\ 0, & Otherwise, \end{cases} \quad (10)$$

*where $\mathcal{N}_i$ denotes the neighboring points in the k-NN graph.*

PROOF. First, we give the Ollivier's Ricci curvature with the mass distribution above and our differentiable formulation as follows

$$Ric^\alpha(i, j) = 1 - \frac{W_1(m_i^\alpha, m_j^\alpha)}{d(\boldsymbol{x}_i, \boldsymbol{x}_j)}, \quad (11)$$

$$Ric^\alpha(i, j) = 1 - \frac{f([\boldsymbol{L}^\alpha \ (\boldsymbol{X} \otimes_\kappa \boldsymbol{W})]_i) - f([\boldsymbol{L}^\alpha \ (\boldsymbol{X} \otimes_\kappa \boldsymbol{W})]_j)}{d(\boldsymbol{x}_i, \boldsymbol{x}_j)}, \quad (12)$$

where we have $f(\boldsymbol{x}) = \boldsymbol{x}\boldsymbol{1}$. Second, we study the relationship between Wasserstein distance and expression as follows,

$$f([\boldsymbol{L}^\alpha \ (\boldsymbol{X} \otimes_\kappa \boldsymbol{W})]_i) - f([\boldsymbol{L}^\alpha \ (\boldsymbol{X} \otimes_\kappa \boldsymbol{W})]_j), \quad (13)$$

where Laplacian matrix $\boldsymbol{L}^\alpha$ takes the form of

$$[\boldsymbol{L}^\alpha]_{ij} = \begin{cases} \alpha, & i = j, \\ (1-\alpha) \frac{1}{D_{ii}}, & [\boldsymbol{A}]_{ij} = 1, \\ 0, & Otherwise, \end{cases} \quad (14)$$

and $\boldsymbol{D}$ is the diagonal degree matrix of the k-NN graph. With Kantorovich-Rubinstein duality [15], Wasserstein distance between two distributions is rewritten as

$$W_1(p, q) = \sup_{\|f\|_L \leq 1} \mathbb{E}_{z \sim p}[f(z)] - \mathbb{E}_{z \sim q}[f(z)], \quad (15)$$

where $f$ is 1–Lipschitz. With Eqs. (10), (13) and (15),

$$\begin{aligned} W_1(m_i^\alpha, m_j^\alpha) &= \sup_{\|f\|_L \leq 1} \sum_{x \in \mathcal{D}} f(x) m_i^\alpha(x) - \sum_{x \in \mathcal{D}} f(x) m_j^\alpha(x) \\ &= \sup_{\|f\|_L \leq 1} [\boldsymbol{L}^\alpha f(\boldsymbol{X})]_i - [\boldsymbol{L}^\alpha f(\boldsymbol{X})]_j, \end{aligned} \quad (16)$$

where $f(\boldsymbol{X}) = (\boldsymbol{X} \otimes_\kappa \boldsymbol{W})\boldsymbol{1}$. The operation of $\otimes_\kappa$ is indeed an affine transform [1], and thus $f$ is 1–Lipschitz with proper scaling according to Cauchy-Schwartz inequality. The supremum holds for any feasible $f$. That is, our differentiable formulation is the upper bound of Ollivier's Ricci curvature. □

# F REPRODUCIBILITY

We specify the network architecture of parametric $v_t$ and $f$ for soft assignment, and further details in the experiment.

On the vector field $v_t$, it is a function over manifold-valued point $\boldsymbol{x}$ and time $t$. First, we perform logarithmic map on $\boldsymbol{x}$ to obtain the projection in Euclidean tangent space. Then, the concatenation of mapped $\boldsymbol{x}$ and time encoding of $t$ is fed into the MLP. We leverage the popular cosine encoding $\phi(t)$ defined as follows,

$$\phi(t) = \sqrt{\frac{1}{d}}[\cos(\omega_1 t + \theta_1), \cos(\omega_2 t + \theta_2), \cdots, \cos(\omega_d t + \theta_d)], \quad (17)$$

where $d$ is the dimension of time encoding, and $\omega$'s and $\theta$'s are parameters. Eq. (17) induces a translation-invariant kernel according to Bochner's theorem. On the soft assignment, the network architecture of $f$ is designed as

$$f(z, \boldsymbol{\pi}, \boldsymbol{\mu}_1, \cdots, \boldsymbol{\mu}_K) = h(\mathrm{Cat}(\boldsymbol{\pi}, \mathrm{Pooling}(z, \boldsymbol{\mu}_1, \cdots, \boldsymbol{\mu}_K))), \quad (18)$$

where Cat denotes vector concatenation. We suggest the mean-pooling, and $h$ is given as MLP. As for dimension reduction, one can leverage an autoencoder in Euclidean space. In RICCINET, we opt for unitizing $\kappa$-right-multiplication given in Table 3, and the weight matrix is jointly learnt with our model. In the discussion, for the variant $\mathbb{M}_0$, we employ the algorithm in [13] to predefine curvature. The algorithm is based on analyzing and enumerating the geodesic triangles. It is time consuming, and thus is expensive for large scale graphs. We set curvature as a learnable parameter of RICCINET. In the training process, learning rate of the optimizer is set as 0.0005, and the dropout of velocity net is set as 0.2 by default.

