# OpenReview forum: "RicciNet: Deep Clustering via A Riemannian Generative Model"
_ACM.org/TheWebConf/2024/Conference — TheWebConf24_

### Official Review · Reviewer_z5FW · 2023-11-20

**Novelty:** 5
**Technical Quality:** 5

**Review:**

This paper proposed a Riemannian generative model for clustering via Ricci flow. The key idea is to encode data point as a Gaussian mixture sample on a Riemannian manifold, and perform geometric adjustments (distance matching and speed matching) based on Ricci curvature during decoding.  Since the discrete optimization will block the gradient backpropagation while computing Ricci curvature, the authors derive a differentiable formulation along with theoretical guarantees. Experimental results show that the proposed model can effectively capture information about complex structures therefore outperforms Euclidean based methods on a variety of datasets.

**Paper Strengths**

1. This paper study deep clustering from a perspective of Riemannian geometry, which can model complex structures more effectively.
2. The idea of using Ricci flow for clustering is interesting.
3. The experiments are comprehensive. The clustering results are obtained after 10 independent runs for each model, and ablation studies are given to show the effectiveness of the proposed components.

**Paper Weaknesses**

1. There are some typos in this paper：
  - In Sec 3.3.2, "With Eqs. (10), (13) and (15)" should be "With Eqs. (10), (13) and (14)".
  - In Sec 4.4, "(1) Euclidean space E, (1) standard hyperbolic manifold" should be "(1) Euclidean space E, (2) standard hyperbolic manifold".
  - In Appendix C Algo.1, "Input: Soft assignment $\sigma$" should be "Input: Soft assignment $a$".
  - In Remark 3, Reference [8] and Reference [27] seem to be the same?
2. There is not much difference between the variant w/oReparameter and the proposed model on USPS dataset. Is it really necessary to replace optimizing the log-likelihood with the loss of differential geometric learning? Will it perform worse on other datasets after replacement? (Please provide comparative results on other datasets.)

**Questions:**

1. Can you provide an intuitive example (e.g., a figure) to illustrate "Ricci flow demonstrates that particles in the manifold tend to aggregate into several submanifolds, influenced by the Ricci curvature" in Sec 1 Introduction?

2. How is Eqs.(6) derived? Is it cited from Reference [27]?

3. The authors say "We follow the geometric intuition thata data point from Gaussian mixture can be expressed as a linear aggregation of the means of component Gaussian in the manifold ...". Please provide more theoretical interpretations.

4. Can you provide a comparative analysis of computational complexity between proposed model and other Euclidean based methods?

**Reviewer Confidence:**

2: The reviewer is willing to defend the evaluation, but it is likely that the reviewer did not understand parts of the paper

**Scope:**

3: The work is somewhat relevant to the Web and to the track, and is of narrow interest to a sub-community

---

### Official Review · Reviewer_yY9a · 2023-11-23

**Novelty:** 6
**Technical Quality:** 5

**Review:**

The study proposes a generative neural Riemannian flow, in which data points move towards their clusters on the manifold, along the shortest Ricci flow. It starts by letting the data be a sample of a Gaussian mixture, then based on the differentiable Ricci curvature from the Riemannian graph convolution, the points move, and cluster.

Some strengths:
- able to cluster complex structures.
- heavily based in geometry.


Some weaknesses:
- small presentation issues.

**Questions:**

1. On the proof of proposition 2, how is equation (8) verified? I think I need some more clarification(perhaps a little more detail) on this proof.
2. Small presentation mistakes: empty ReadMe. typo for neural in line 258.

**Reviewer Confidence:**

2: The reviewer is willing to defend the evaluation, but it is likely that the reviewer did not understand parts of the paper

**Scope:**

3: The work is somewhat relevant to the Web and to the track, and is of narrow interest to a sub-community

---

### Official Review · Reviewer_AtKF · 2023-11-25

**Novelty:** 5
**Technical Quality:** 5

**Review:**

This paper proposes a novel Riemannian generative model, which aims to solve the shortcomings of traditional deep clustering methods in complex structures. Experimental results prove that RicciNet outperforms Euclidean deep methods. This paper is well written and logically clear. The article focuses on how to solve several problems of Ricci flow to introduce it into deep clustering. However, the answer to why Ricci flow is suitable for deep clustering is illegible. And there are still several doubts in the experimental part.

**Questions:**

(1)	In the introduction section, the explanation of the suitability of Ricci flow for deep clustering appears to be too brief.

(2)	The proposed method parameterized by a Riemannian graph convolution on the k-NN graph. Why should the kNN graph be chosen? Are there any other better options?

(3)	The motivation of this article is to solve the shortcomings of clustering under complex structures. However, previous work has used non-Euclidean embeddings to solve this problem, such as kernel clustering. It is recommended to illustrate the shortcomings of previous work and highlight the benefits of using Ricci flow.

(4)	In the experimental section, it appears that the geometric trick yields better results with small datasets. Could you provide an explanation for this observation?

(5)	Why was the curvature setting experiment with $\mathbb{M}_{0}$ performed exclusively on Cora and Citeseer?

(6)	To investigate the influence of initial parameters on the results, it might be beneficial to include a parameter sensitivity analysis in the experimental section.

**Reviewer Confidence:**

4: The reviewer is certain that the evaluation is correct and very familiar with the relevant literature

**Scope:**

3: The work is somewhat relevant to the Web and to the track, and is of narrow interest to a sub-community

---

### Official Review · Reviewer_8iwZ · 2023-11-25

**Novelty:** 5
**Technical Quality:** 4

**Review:**

**Summary**

This paper introduces a new model called RicciNet, a Riemannian generative model that incorporates a dynamic self-clustering process using Ricci flow. The model represents data points' movement in the manifold influenced by Ricci curvatures, characterized by a parametric velocity described by an Ordinary Differential Equation. The encoding involves Gaussian mixture samples with reparameterization approaches. The model includes a differentiable Ricci curvature and a geometric learning approach that studies the trajectory's regularity through distance and velocity matching.

**Strengths:**
1. The motivation of this paper is clear as the Euclidean-based clustering methods fail to capture some complex structures.
2. The proposed method has a theoretical guarantee.
3. The authors conduct extensive experiments to demonstrate the effectiveness of the proposed method.

**Weaknesses:**
1. The experiment on graph data might not be convincing. Graph data is different from other types of data (e.g., image) due to its non-iid nature. The baseline methods could not capture the topological information of the graph and there exist hyperbolic graph neural networks [1][2], which aim to learn the node embedding in the hyperbolic space. To better demonstrate the effectiveness of the proposed method for graph data, it's recommended to compare it with some graph baseline methods, specifically [1], [2].
2. The paper does not include the parameter analysis for $\beta$, which is used to balance two terms in equation 21. In addition, the authors do not specify the search space of $\beta$.
3. The authors visualize the clustering results on two artificial datasets. The visualization of real-world datasets might be more convincing.

[1] Liu, Qi, Maximilian Nickel, and Douwe Kiela. "Hyperbolic graph neural networks." Advances in neural information processing systems 32 (2019).
[2] Chami, Ines, Zhitao Ying, Christopher Ré, and Jure Leskovec. "Hyperbolic graph convolutional neural networks." Advances in neural information processing systems 32 (2019).

**Questions:**

Q1: For two graph datasets used in the experiment, do you use the adjacency matrix via the kNN method or the raw adjacency matrix in the graph dataset for the proposed method (i.e., equation 12)?

Q2: This paper introduces two variants of RicciNets with either Gumble parameterization or geometric trick. Table 1 shows that one variant outperforms another in some datasets but fails to do so in the rest datasets. Given a dataset, which method should be used for evaluation?

**Reviewer Confidence:**

2: The reviewer is willing to defend the evaluation, but it is likely that the reviewer did not understand parts of the paper

**Scope:**

2: The connection to the Web is incidental, e.g., use of Web data or API

---

### Decision · Program_Chairs · 2024-01-22

**Decision:**

Accept

**Comment:**

This paper considers the problem of deep clustering in Riemannian geometry, introducing a novel generative model.

 Reviewers found that the paper was well-written and provided a clear motivation to cluster some complex structures not handled by Euclidean methods. The new methods provide both theoretical and practical benefits.

 While the reviewers asked a number of questions, most were not drawbacks but rather additional curiosities. An exception was that multiple reviewers wanted additional explanation/visualization in the introduction when explaining why Ricci flow is suitable for deep clustering. In general, though, the authors have provided useful responses, for which the reviewers are grateful.

 In summary, the authors have addressed a number of challenges to propose a novel Riemannian generative model with a theoretical guarantee and practical value for certain complex patterns. In rebuttal, it is also shown to be efficient.